# Multi-task Representation Learning for Pure Exploration in Bilinear Bandits

**Subhojyoti Mukherjee**
ECE Department
UW-Madison
Wisconsin, Madison
smukherjee27@wisc.edu

Qiaomin Xie
ISyE Department
UW-Madison
Wisconsin, Madison

Josiah P. Hanna
CS Department
UW-Madison
Wisconsin, Madison

Robert Nowak
ECE Department
UW-Madison
Wisconsin, Madison

## Abstract

We study multi-task representation learning for the problem of pure exploration in bilinear bandits. In bilinear bandits, an action takes the form of a pair of arms from two different entity types and the reward is a bilinear function of the known feature vectors of the arms. In the *multi-task bilinear bandit problem*, we aim to find optimal actions for multiple tasks that share a common low-dimensional linear representation. The objective is to leverage this characteristic to expedite the process of identifying the best pair of arms for all tasks. We propose the algorithm GOBLIN that uses an experimental design approach to optimize sample allocations for learning the global representation as well as minimize the number of samples needed to identify the optimal pair of arms in individual tasks. To the best of our knowledge, this is the first study to give sample complexity analysis for pure exploration in bilinear bandits with shared representation. Our results demonstrate that by learning the shared representation across tasks, we achieve significantly improved sample complexity compared to the traditional approach of solving tasks independently.

## 1 Introduction

Bilinear bandits (Jun et al., 2019; Lu et al., 2021; Kang et al., 2022) are an important class of sequential decision-making problems. In bilinear bandits (as opposed to the standard linear bandit setting) we are given a pair of arms $\mathbf{x}_t \in \mathbb{R}^{d_1}$ and $\mathbf{z}_t \in \mathbb{R}^{d_2}$ at every round $t$ and the interaction of this pair of arms with a low-rank hidden parameter, $\boldsymbol{\Theta}_* \in \mathbb{R}^{d_1 \times d_2}$ generates the noisy feedback (reward) $r_t = \mathbf{x}_t^\top \boldsymbol{\Theta}_* \mathbf{z}_t + \eta_t$. The $\eta_t$ is random 1-subGaussian noise.

A lot of real-world applications exhibit the above bilinear feedback structure, particularly applications that involve selecting pairs of items and evaluating their compatibility. For example, in a drug discovery application, scientists may want to determine whether a particular (drug, protein) pair interacts in the desired way (Luo et al., 2017). Likewise, an online dating service might match a pair of people and gather feedback about their compatibility (Shen et al., 2023). A clothing website's recommendation system may suggest a pair of items (top, bottom) for a customer based on their likelihood of matching (Reyes et al., 2021). In all of these scenarios, the two items are considered as a single unit, and the system must utilize available feature vectors $(\mathbf{x}_t, \mathbf{z}_t)$ to learn which features of the pairs are most indicative of positive feedback in order to make effective recommendations. All the previous works in this setting (Jun et al., 2019; Lu et al., 2021; Kang et al., 2022) exclusively

37th Conference on Neural Information Processing Systems (NeurIPS 2023).

focused on maximizing the number of pairs with desired interactions discovered over time (regret minimization). However, in many real-world applications where obtaining a sample is expensive and time-consuming, e.g., clinical trials (Zhao et al., 2009; Zhang et al., 2012), it is often desirable to identify the optimal option using as few samples as possible, i.e., we face the pure exploration scenario (Fiez et al., 2019; Katz-Samuels et al., 2020) rather than regret minimization.

Moreover, in various decision-making scenarios, we may encounter multiple interrelated tasks such as treatment planning for different diseases (Bragman et al., 2018) and content optimization for multiple websites (Agarwal et al., 2009). Often, there exists a shared representation among these tasks, such as the features of drugs or the representations of website items. Therefore, we can leverage this shared representation to accelerate learning. This area of research is called multi-task representation learning and has recently generated a lot of attention in machine learning (Bengio et al., 2013; Li et al., 2014; Maurer et al., 2016; Du et al., 2020; Tripuraneni et al., 2021). There are many applications of this multi-task representation learning in real-world settings. For instance, in clinical treatment planning, we seek to determine the optimal treatments for multiple diseases, and there may exist a low-dimensional representation common to multiple diseases. To avoid the time-consuming process of conducting clinical trials for individual tasks and collecting samples, we utilize the shared representation and decrease the number of required samples.

The above multi-task representation learning naturally shows up in bilinear bandit setting as follows: Let there be $M$ tasks indexed as $m = 1, 2, \ldots, M$ with each task having its own hidden parameter $\boldsymbol{\Theta}_{m,*} \in \mathbb{R}^{d_1 \times d_2}$. Let each $\boldsymbol{\Theta}_{m,*}$ has a decomposition of $\boldsymbol{\Theta}_{m,*} = \mathbf{B}_1 \mathbf{S}_{m,*} \mathbf{B}_2^\top$, where $\mathbf{B}_1 \in \mathbb{R}^{d_1 \times k_1}$ and $\mathbf{B}_2 \in \mathbb{R}^{d_2 \times k_2}$ are shared across tasks, but $\mathbf{S}_{m,*} \in \mathbb{R}^{k_1 \times k_2}$ is specific for task $m$. We assume that $k_1, k_2 \ll d_1, d_2$ and $M \gg d_1, d_2$. Thus, $\mathbf{B}_1$ and $\mathbf{B}_2$ provide a means of dimensionality reduction. Furthermore, we assume that each $\mathbf{S}_{m,*}$ has rank $r \ll \min\{k_1, k_2\}$. In the terminology of multi-task representation learning $\mathbf{B}_1, \mathbf{B}_2$ are called *feature extractors* and $\mathbf{x}_{m,t}, \mathbf{z}_{m,t}$ are called *rich observations* (Yang et al., 2020, 2022; Du et al., 2023). The reward for the task $m \in \{1, 2, \ldots, M\}$ at round $t$ is

$$r_{m,t} = \mathbf{x}_{m,t}^\top \boldsymbol{\Theta}_{m,*} \mathbf{z}_{m,t} + \eta_{m,t} = \underbrace{\mathbf{x}_{m,t}^\top \mathbf{B}_1}_{\mathbf{g}_{m,t}^\top} \mathbf{S}_{m,*} \underbrace{\mathbf{B}_2^\top \mathbf{z}_{m,t}}_{\mathbf{v}_{m,t}} + \eta_{m,t} = \mathbf{g}_{m,t}^\top \mathbf{S}_{m,*} \mathbf{v}_{m,t} + \eta_{m,t}. \quad (1)$$

Observe that similar to the learning procedure in Yang et al. (2020, 2022), at each round $t = 1, 2, \cdots$, for each task $m \in [M]$, the learner selects a left and right action $\mathbf{x}_{m,t} \in \mathcal{X}$ and $\mathbf{z}_{m,t} \in \mathcal{Z}$. After the player commits the batch of actions for each task $\{\mathbf{x}_{m,t}, \mathbf{z}_{m,t} : m \in [M]\}$, it receives the batch of rewards $\{r_{m,t} : m \in [M]\}$. Also note that in (1) we define the $\widetilde{\mathbf{g}}_{m,t} \in \mathbb{R}^{k_1}, \widetilde{\mathbf{v}}_{m,t} \in \mathbb{R}^{k_2}$ as the latent features, and both $\widetilde{\mathbf{g}}_{m,t}, \widetilde{\mathbf{v}}_{m,t}$ are unknown to the learner and needs to be learned for each task $m$ (hence the name multi-task representation learning).

In this paper, we focus on pure exploration for multi-task representation learning in bilinear bandits where the goal is to find the optimal left arm $\mathbf{x}_{m,*}$ and right arm $\mathbf{z}_{m,*}$ for each task $m$ with a minimum number of samples (fixed confidence setting). First, consider a single-task setting and let $\boldsymbol{\Theta}_*$ have low rank $r$. Let the SVD of the $\boldsymbol{\Theta}_* = \mathbf{U} \mathbf{D} \mathbf{V}^\top$. Prima-facie, if $\mathbf{U}$ and $\mathbf{V}$ are known then one might want to project all the left and right arms in the $r \times r$ subspace of $\mathbf{U}$ and $\mathbf{V}$ and reduce the bilinear bandit problem into a $r^2$ dimension linear bandit setting. Then one can apply one of the algorithms from Soare et al. (2014); Fiez et al. (2019); Katz-Samuels et al. (2020) to solve this $r^2$ dimensional linear bandit pure exploration problem. Following the analysis of this line of work (in linear bandits) (Mason et al., 2021; Mukherjee et al., 2022, 2023) one might conjecture that a sample complexity bound of $\widetilde{O}(r^2/\Delta^2)$ is possible where $\Delta$ is the minimum reward gap and $\widetilde{O}(\cdot)$ hides log factors. Similarly, for the multi-task setting one might be tempted to use the linear bandit analysis of Du et al. (2023) to convert this problem into $M$ concurrent $r^2$ dimensional linear bandit problems with shared representation and achieve a sample complexity bound of $\widetilde{O}(Mr^2/\Delta^2)$. However, these matrices (subspaces) are unknown and so there is a model mismatch as noted in the regret analysis of bilinear bandits (Jun et al., 2019; Lu et al., 2021; Kang et al., 2022). Thus it is difficult to apply the $r^2$ dimensional linear bandit sample complexity analysis. Following the regret analysis of bilinear bandit setting by Jun et al. (2019); Lu et al. (2021); Kang et al. (2022) we know that the effective dimension is actually $(d_1 + d_2)r$. Similarly for the multi-task representation learning the effective dimension should scale with the learned latent features $(k_1 + k_2)r$. Hence the natural questions to ask are these:

> **1) Can we design a single-task pure exploration bilinear bandit algorithm whose sample complexity scales as $\widetilde{O}((d_1 + d_2)r/\Delta^2)$?**

**2) Can we design an algorithm for multi-task pure exploration bilinear bandit problem that can learn the latent features and has sample complexity that scales as $\widetilde{O}(M(k_1 + k_2)r/\Delta^2)$?**

In this paper, we answer both these questions affirmatively. In doing so, we make the following novel contributions to the growing literature of multi-task representation learning in online settings:

**1)** We formulate the multi-task bilinear representation learning problem. To our knowledge, this is the first work that explores pure exploration in a multi-task bilinear representation learning setting.

**2)** We proposed the algorithm GOBLIN for a single-task pure exploration bilinear bandit setting whose sample complexity scales as $\widetilde{O}((d_1 + d_2)r/\Delta^2)$. This improves over RAGE (Fiez et al., 2019) whose sample complexity scales as $\widetilde{O}((d_1 d_2)/\Delta^2)$.

**3)** Our algorithm GOBLIN for multi-task pure exploration bilinear bandit problem learns the latent features and has sample complexity that scales as $\widetilde{O}(M(k_1 + k_2)r/\Delta^2)$. This improves over DouExpDes (Du et al., 2023) whose samples complexity scales as $\widetilde{O}(M(k_1 k_2)/\Delta^2)$.

**Preliminaries:** We assume that $\|\mathbf{x}\|_2 \leq 1$, $\|\mathbf{z}\|_2 \leq 1$, $\|\boldsymbol{\Theta}_*\|_F \leq S_0$ and the $r$-th largest singular value of $\boldsymbol{\Theta}_* \in \mathbb{R}^{d_1 \times d_2}$ is $S_r$. Let $p := d_1 d_2$ denote the ambient dimension, and $k = (d_1 + d_2)r$ denote the effective dimension. Let $[n] := \{1, 2, \ldots, n\}$. Let $\mathbf{x}_*, \mathbf{z}_* := \arg\max_{\mathbf{x}, \mathbf{z}} \mathbf{x}^\top \boldsymbol{\Theta}_* \mathbf{z}$. For any $\mathbf{x}, \mathbf{z}$ define the gap $\Delta(\mathbf{x}, \mathbf{z}) := \mathbf{x}_*^\top \boldsymbol{\Theta}_* \mathbf{z}_* - \mathbf{x}^\top \boldsymbol{\Theta}_* \mathbf{z}$ and furthermore $\Delta = \min_{\mathbf{x} \neq \mathbf{x}_*, \mathbf{z} \neq \mathbf{z}_*} \Delta(\mathbf{x}, \mathbf{z})$. Similarly, for any arbitrary vector $\mathbf{w} \in \mathcal{W}$ define the gap of $\mathbf{w} \in \mathbb{R}^p$ as $\Delta(\mathbf{w}) := (\mathbf{w}_* - \mathbf{w})^\top \boldsymbol{\theta}_*$, for some $\boldsymbol{\theta}_* \in \mathbb{R}^p$ and furthermore, $\Delta = \min_{\mathbf{w} \neq \mathbf{w}_*} \Delta(\mathbf{w})$. If $\mathbf{A} \in \mathbb{R}_{\geq 0}^{d \times d}$ is a positive semidefinite matrix, and $\mathbf{w} \in \mathbb{R}^p$ is a vector, let $\|\mathbf{w}\|_{\mathbf{A}}^2 := \mathbf{w}^\top \mathbf{A} \mathbf{w}$ denote the induced semi-norm. Given any vector $\mathbf{b} \in \mathbb{R}^{|\mathcal{W}|}$ we denote the $\mathbf{w}$-th component as $\mathbf{b_w}$. Let $\Delta_{\mathcal{W}} := \left\{ \mathbf{b} \in \mathbb{R}^{|\mathcal{W}|} : \mathbf{b_w} \geq 0, \sum_{\mathbf{w} \in \mathcal{W}} \mathbf{b_w} = 1 \right\}$ denote the set of probability distributions on $\mathcal{W}$. We define $\mathcal{Y}(\mathcal{W}) = \{\mathbf{w} - \mathbf{w}' : \forall \mathbf{w}, \mathbf{w}' \in \mathcal{W}, \mathbf{w} \neq \mathbf{w}'\}$ as the directions obtained from the differences between each pair of arms and $\mathcal{Y}^*(\mathcal{W}) = \{\mathbf{w}_* - \mathbf{w} : \forall \mathbf{w} \in \mathcal{W} \backslash \mathbf{w}_*\}$ as the directions obtained from the differences between the optimal arm and each suboptimal arm.

## 2 Pure Exploration in Single-Task Bilinear Bandits

In this section, we consider pure exploration in a single-task bilinear bandit setting as a warm-up to the main goal of learning representations for the multi-task bilinear bandit. To our knowledge, this is the first study of pure exploration in single-task bilinear bandits. We first recall the single-task bilinear bandit setting as follows: At every round $t = 1, 2, \ldots$ the learner observes the reward $r_t = \mathbf{x}_t^\top \boldsymbol{\Theta}_* \mathbf{z}_t + \eta_t$ where the low rank hidden parameter $\boldsymbol{\Theta}_* \in \mathbb{R}^{d_1 \times d_2}$ is unknown to the learner, $\mathbf{x}_t \in \mathbb{R}^{d_1}$, $\mathbf{z}_t \in \mathbb{R}^{d_2}$ are visible to the learner, and $\eta_t$ is a 1-sub-Gaussian noise. We assume that the matrix $\boldsymbol{\Theta}_*$ has a low rank $r$ which is known to the learner and $d_1, d_2 \gg r$. Finally recall that the goal is to identify the optimal left and right arms $\mathbf{x}_*, \mathbf{z}_*$ with a minimum number of samples.

We propose a phase-based, two-stage arm elimination algorithm called **G-O**ptimal Design for **Bilin**ear Bandits (abbreviated as GOBLIN). GOBLIN proceeds in phases indexed by $\ell = 1, 2, \ldots$ As this is a pure-exploration problem, the total number of samples is controlled by the total phases which depends on the intrinsic problem complexity. Each phase $\ell$ of GOBLIN consists of two stages; the estimation of $\boldsymbol{\Theta}_*$ stage, which runs for $\tau_\ell^E$ rounds, and pure exploration in rotated arms stage that runs for $\tau_\ell^G$ rounds. We will define $\tau_\ell^E$ in Section 2.1, while rotated arms and $\tau_\ell^G$ are defined in Section 2.2. At the end of every phase, GOBLIN eliminates sub-optimal arms to build the active set for the next phase and stops when only the optimal left and right arms are remaining. Now we discuss the individual stages that occur at every phase $\ell$ of GOBLIN.

### 2.1 Estimating Subspaces of $\boldsymbol{\Theta}_*$ (Stage 1 of the $\ell$-th phase)

In the first stage of phase $\ell$, GOBLIN estimates the row and column sub-spaces $\boldsymbol{\Theta}_*$. Then GOBLIN uses these estimates to reduce the bilinear bandit problem in the original ambient dimension $p := d_1 d_2$ to a lower effective dimension $k := (d_1 + d_2)r$. To do this, GOBLIN first vectorizes the $\mathbf{x} \in \mathbb{R}^{d_1}$, $\mathbf{z} \in \mathbb{R}^{d_2}$ into a new vector $\overline{\mathbf{w}} \in \mathbb{R}^p$ and then solves the $E$-optimal design in Step 3 of

Algorithm 1 (Pukelsheim, 2006; Jun et al., 2019; Du et al., 2023). Let the solution to the $E$-optimal design problem at the stage 1 of $\ell$-th phase be denoted by $\mathbf{b}_\ell^E$. Then GOBLIN samples each $\overline{\mathbf{w}}$ for $\lceil \tau_\ell^E \mathbf{b}_{\ell,\overline{\mathbf{w}}}^E \rceil$ times, where $\tau_\ell^E = \widetilde{O}(\sqrt{d_1 d_2 r}/S_r)$ (step 7 of Algorithm 1). In this paper, we sample an arm $\lceil \tau_\ell^E \mathbf{b}_{\ell,\overline{\mathbf{w}}}^E \rceil$ number of times. However, this may lead to over-sampling of an arm than what the design ($G$ or $E$-optimal) is actually suggesting. However, we can match the number of allocations of an arm to the design using an *efficient Rounding Procedures* (see Pukelsheim (2006); Fiez et al. (2019)). Let $\widehat{\boldsymbol{\Theta}}_\ell$ be estimate of $\boldsymbol{\Theta}_*$ in stage 1 of phase $\ell$. GOBLIN estimates this by solving the following well-defined regularized minimization problem with nuclear norm penalty:

$$\widehat{\boldsymbol{\Theta}}_\ell = \underset{\boldsymbol{\Theta} \in \mathbb{R}^{d_1 \times d_2}}{\arg\min} \ L_\ell(\boldsymbol{\Theta}) + \gamma_\ell \|\boldsymbol{\Theta}\|_{\text{nuc}}, \quad L_\ell(\boldsymbol{\Theta}) = \langle \boldsymbol{\Theta}, \boldsymbol{\Theta} \rangle - \frac{2}{\tau_\ell^E} \sum_{s=1}^{\tau_\ell^E} \langle \widetilde{\psi}_\nu (r_s \cdot Q(\mathbf{x}_s \mathbf{z}_s^\top)), \boldsymbol{\Theta} \rangle \quad (2)$$

where $Q(\cdot)$, $\widetilde{\psi}_\nu(\cdot)$, are appropriate functions stated in Definition 1, 3 respectively in Appendix A.3. The $Q(\cdot)$ function takes as input the rank-one matrix $\mathbf{x}_s \mathbf{z}_s^\top$ which is obtained after reshaping $\overline{\mathbf{w}}_s$. Note that $\mathbf{x}_s$, and $\mathbf{z}_s$ are the observed vectors in $d_1$ and $d_2$ dimension and $\widehat{\boldsymbol{\Theta}}_\ell \in \mathbb{R}^{d_1 \times d_2}$ Finally, set the regularization parameter $\gamma_\ell := 4 \sqrt{\frac{2(4+S_0^2)C d_1 d_2 \log(2(d_1+d_2)/\delta)}{\tau_\ell^E}}$. This is in step 8 of Algorithm 1.

## 2.2 Optimal Design for Rotated Arms (Stage 2 of $\ell$-th phase)

In stage 2 of phase $\ell$, GOBLIN leverages the information about the learned sub-space of $\boldsymbol{\Theta}_*$ to rotate the arm set and then run the optimal design on the rotated arm set. Once we recover $\widehat{\boldsymbol{\Theta}}_\ell$, one might be tempted to run a pure exploration algorithm (Soare et al., 2014; Fiez et al., 2019; Katz-Samuels et al., 2020; Zhu et al., 2021) to identify $\mathbf{x}_*$ and $\mathbf{z}_*$. However, then the sample complexity will scale with $d_1 d_2$. In contrast GOBLIN uses the information about the learned sub-space of $\boldsymbol{\Theta}_*$ to reduce the problem from ambient dimension $d_1 d_2$ to effective dimension $(d_1 + d_2)r$. This reduction is done as follows: Let $\widehat{\boldsymbol{\Theta}}_\ell = \widehat{\mathbf{U}}_\ell \widehat{\mathbf{D}}_\ell \widehat{\mathbf{V}}_\ell^\top$ be the SVD of $\widehat{\boldsymbol{\Theta}}_\ell$ in the $\ell$-th phase. Let $\widehat{\mathbf{U}}_\perp^\ell$ and $\widehat{\mathbf{V}}_\perp^\ell$ be orthonormal bases of the complementary subspaces of $\widehat{\mathbf{U}}_\ell$ and $\widehat{\mathbf{V}}_\ell$ respectively. Let $\mathcal{X}_\ell$ and $\mathcal{Z}_\ell$ be the active set of arms in the stage 2 of phase $\ell$. Then rotate the arm sets such that new rotated arm sets are as follows:

$$\underline{\mathcal{X}}_\ell = \{\underline{\mathbf{x}} = [\widehat{\mathbf{U}}_\ell \widehat{\mathbf{U}}_\ell^\perp]^\top \mathbf{x} \mid \mathbf{x} \in \mathcal{X}_\ell\}, \ \underline{\mathcal{Z}}_\ell = \{\underline{\mathbf{z}} = [\widehat{\mathbf{V}}_\ell \widehat{\mathbf{V}}_\ell^\perp]^\top \mathbf{z} \mid \mathbf{z} \in \mathcal{Z}_\ell\}. \quad (3)$$

Let $\widehat{\mathbf{H}}_\ell = [\widehat{\mathbf{U}}_\ell \widehat{\mathbf{U}}_\ell^\perp]^\top \widehat{\boldsymbol{\Theta}}_\ell [\widehat{\mathbf{V}}_\ell \widehat{\mathbf{V}}_\ell^\perp]$. Then define vectorized arm set so that the last $(d_1 - r) \cdot (d_2 - r)$ components are from the complementary subspaces as follows:

$$\underline{\mathcal{W}}_\ell = \big\{ \big[ \mathbf{vec}\left(\underline{\mathbf{x}}_{1:r} \underline{\mathbf{z}}_{1:r}^\top\right); \mathbf{vec}\left(\underline{\mathbf{x}}_{r+1:d_1} \underline{\mathbf{z}}_{1:r}^\top\right); \mathbf{vec}\left(\underline{\mathbf{x}}_{1:r} \underline{\mathbf{z}}_{r+1:d_2}^\top\right);$$
$$\mathbf{vec}\left(\underline{\mathbf{x}}_{r+1:d_1} \underline{\mathbf{z}}_{r+1:d_2}^\top\right) \big] \in \mathbb{R}^{d_1 d_2} : \underline{\mathbf{x}} \in \mathcal{X}_\ell, \underline{\mathbf{z}} \in \mathcal{Z}_\ell \big\}$$
$$\widehat{\boldsymbol{\theta}}_{\ell,1:k} = [\mathbf{vec}(\widehat{\mathbf{H}}_{\ell,1:r,1:r}); \mathbf{vec}(\widehat{\mathbf{H}}_{\ell,r+1:d_1,1:r}); \mathbf{vec}(\widehat{\mathbf{H}}_{\ell,1:r,r+1:d_2})],$$
$$\widehat{\boldsymbol{\theta}}_{\ell,k+1:p} = \mathbf{vec}(\widehat{\mathbf{H}}_{\ell,r+1:d_1,r+1:d_2}). \quad (4)$$

which implies $\|\widehat{\boldsymbol{\theta}}_{k+1:p}\|_2 = O\left(d_1 d_2 r/\tau_\ell^E\right)$ by Lemma 3 in Appendix A.1. So the last $p - k$ components of $\widehat{\boldsymbol{\theta}}_\ell$ are very small compared to the first $k$ components. Hence, GOBLIN has now reduced the $d_1 d_2$ dimensional linear bandit to $(d_1 + d_2)r$ dimensional linear bandit using (3), (4). This is shown in step 10 of Algorithm 1.

Now in stage 2 of phase $\ell$, GOBLIN implements $G$-optimal design (Pukelsheim, 2006; Fiez et al., 2019) in the rotated arm set $\underline{\mathcal{X}}_\ell, \underline{\mathcal{Z}}_\ell$ defined in (3). To do this, first GOBLIN defines the rotated vector $\underline{\mathbf{w}} = [\underline{\mathbf{x}}_{1:d_1}; \underline{\mathbf{z}}_{1:d_2}] \in \mathbb{R}^p$ that belong to the set $\underline{\mathcal{W}}_\ell$. Then GOBLIN solves the $G$-optimal design (Pukelsheim, 2006) as follows:

$$\widehat{\mathbf{b}}_\ell^G = \underset{\mathbf{b}_{\underline{\mathbf{w}}}}{\arg\min} \ \max_{\underline{\mathbf{w}}, \underline{\mathbf{w}}' \in \underline{\mathcal{W}}_\ell} \|\underline{\mathbf{w}} - \underline{\mathbf{w}}'\|_{(\sum_{\underline{\mathbf{w}} \in \underline{\mathcal{W}}} \mathbf{b}_{\underline{\mathbf{w}}} \underline{\mathbf{w}} \underline{\mathbf{w}}^\top + \boldsymbol{\Lambda}_\ell/n)^{-1}}^2. \quad (5)$$

This is shown in step 11 of Algorithm 1 and $\boldsymbol{\Lambda}_\ell$ is defined in (6). It can be shown that sampling according to $\widehat{\mathbf{b}}_\ell^G$ leads to the optimal sample complexity. This is discussed in Remark 1 in Appendix A.2. The key point to note from (5) is that due to the estimation in the rotated arm space $\underline{\mathcal{W}}_\ell$ we are guaranteed that the support of $\mathbf{supp}(\widehat{\mathbf{b}}_\ell^G) \leq \widetilde{O}(k(k+1)/2)$ (Pukelsheim, 2006). On the other hand,

if the G-optimal design of Fiez et al. (2019); Katz-Samuels et al. (2020) are run in $d_1 d_2$ dimension then the support of $\widehat{\mathbf{b}}_\ell^G$ will scale with $d_1 d_2$ which will lead to higher sample complexity. Then GOBLIN samples each $\underline{\mathbf{w}} \in \underline{\mathcal{W}}_\ell$ for $\lceil \tau_\ell^G \mathbf{b}_{\ell,\underline{\mathbf{w}}}^G \rceil$ times, where $\tau_\ell^G := \lceil \frac{8B_*^\ell \rho^G(\mathcal{Y}(\mathcal{W}_\ell)) \log(4\ell^2|\mathcal{W}|/\delta)}{\epsilon_\ell^2} \rceil$. Note that the total length of phase $\ell$, combining stages 1 and 2 is $(\tau_\ell^E + \tau_\ell^G)$ rounds. Observe that the stage 1 design is on the whole arm set $\overline{\mathcal{W}}$ whereas stage 2 design is on the refined active set $\underline{\mathcal{W}}_\ell$.

Let the observed features in stage 2 of phase $\ell$ be denoted by $\underline{\mathbf{W}}_\ell \in \mathbb{R}^{\tau_\ell^G \times p}$, and $\mathbf{r}_\ell \in \mathbb{R}^{\tau_\ell^G}$ be the observed rewards. Define the diagonal matrix $\mathbf{\Lambda}_\ell$ as

$$\mathbf{\Lambda}_\ell = \mathbf{diag}[\underbrace{\lambda, \dots, \lambda}_{k}, \underbrace{\lambda_\ell^\perp, \dots, \lambda_\ell^\perp}_{p-k}] \tag{6}$$

where, $\lambda_\ell^\perp := \tau_{\ell-1}^G/8k \log(1 + \tau_{\ell-1}^G/\lambda) \gg \lambda$. Deviating from Soare et al. (2014); Fiez et al. (2019) GOBLIN constructs a regularized least square estimator at phase $\ell$ as follows

$$\widehat{\boldsymbol{\theta}}_\ell = \arg\min_{\boldsymbol{\theta} \in \mathbb{R}^p} \frac{1}{2} \|\underline{\mathbf{W}}_\ell \boldsymbol{\theta} - \mathbf{r}_\ell\|_2^2 + \frac{1}{2}\|\boldsymbol{\theta}\|_{\mathbf{\Lambda}_\ell}^2. \tag{7}$$

This regularized least square estimator in (7) forces the last $p - k$ components of $\widehat{\boldsymbol{\theta}}_\ell$ to be very small compared to the first $k$ components. Then GOBLIN builds the estimate $\widehat{\boldsymbol{\theta}}_\ell$ from (7) only from the observations from this phase (step 13 in Algorithm 1) and eliminates sub-optimal actions in step 14 in Algorithm 1 using the estimator $\widehat{\boldsymbol{\theta}}_\ell$. Finally GOBLIN eliminates sub-optimal arms to build the next phase active set $\underline{\mathcal{W}}_\ell$ and stops when $|\underline{\mathcal{W}}_\ell| = 1$. GOBLIN outputs the arm in $\underline{\mathcal{W}}_\ell$ and reshapes it to get the $\widehat{\mathbf{x}}_*$ and $\widehat{\mathbf{z}}_*$. The full pseudocode is presented in Algorithm 1.

---

**Algorithm 1** G-Optimal Design for Bilinear Bandits (GOBLIN) for single-task setting

---

1: **Input:** arm set $\mathcal{X}, \mathcal{Z}$, confidence $\delta$, rank $r$ of $\mathbf{\Theta}_*$, spectral bound $S_r$ of $\mathbf{\Theta}_*$, $S, S_\ell^\perp :=$ $\frac{8d_1 d_2 r}{\tau_\ell^E S_r^2} \log\left(\frac{d_1+d_2}{\delta_\ell}\right)$, $\lambda, \lambda_\ell^\perp := \tau_{\ell-1}^G/8(d_1+d_2)r \log(1 + \frac{\tau_{\ell-1}^G}{\lambda})$. Let $p := d_1 d_2$, $k := (d_1+d_2)r$.

2: Let $\underline{\mathcal{W}}_1 \leftarrow \underline{\mathcal{W}}, \ell \leftarrow 1$, $\tau_0^G := \log(4\ell^2|\mathcal{X}|/\delta)$. Define $\mathbf{\Lambda}_\ell$ as in (6), $B_*^\ell := (8\sqrt{\lambda}S + \sqrt{\lambda_\ell^\perp S_\ell^\perp})$.

3: Define a vectorized arm $\overline{\mathbf{w}} := [\mathbf{x}_{1:d_1}; \mathbf{z}_{1:d_2}]$ and $\overline{\mathbf{w}} \in \overline{\mathcal{W}}$. Let $\tau_\ell^E := \frac{\sqrt{8d_1 d_2 r \log(4\ell^2|\mathcal{W}|/\delta_\ell)}}{S_r}$. Let the E-optimal design be $\mathbf{b}_\ell^E := \arg\min_{\mathbf{b} \in \triangle_{\overline{\mathcal{W}}}} \|(\sum_{\overline{\mathbf{w}} \in \overline{\mathcal{W}}} \mathbf{b}_{\overline{\mathbf{w}}} \overline{\mathbf{w}}\, \overline{\mathbf{w}}^\top)^{-1}\|$.

4: **while** $|\underline{\mathcal{W}}_\ell| > 1$ **do**

5:     $\epsilon_\ell = 2^{-\ell}$, $\delta_\ell = \delta/\ell^2$.

6:     **(Stage 1:) Explore the Low-Rank Subspace**

7:     Pull arm $\overline{\mathbf{w}} \in \overline{\mathcal{W}}$ exactly $\lceil \widehat{\mathbf{b}}_{\ell,\overline{\mathbf{w}}}^E \tau_\ell^E \rceil$ times and observe rewards $r_t$, for $t = 1, \dots, \tau_\ell^E$.

8:     Compute $\widehat{\mathbf{\Theta}}_\ell$ using (2).

9:     **(Stage 2:) Reduction to low dimensional linear bandits**

10:     Let the SVD of $\widehat{\mathbf{\Theta}}_\ell = \widehat{\mathbf{U}}_\ell \widehat{\mathbf{D}}_\ell \widehat{\mathbf{V}}_\ell^\top$. Rotate arms in active set $\underline{\mathcal{W}}_{\ell-1}$ to build $\underline{\mathcal{W}}_\ell$ following (4).

11:     Let $\widehat{\mathbf{b}}_\ell^G := \arg\min_{\mathbf{b}_{\underline{\mathbf{w}}}} \max_{\underline{\mathbf{w}},\underline{\mathbf{w}}' \in \underline{\mathcal{W}}_\ell} \|\underline{\mathbf{w}} - \underline{\mathbf{w}}'\|_{(\sum_{\underline{\mathbf{w}} \in \underline{\mathcal{W}}} \mathbf{b}_{\underline{\mathbf{w}}} \underline{\mathbf{w}}\, \underline{\mathbf{w}}^\top + \mathbf{\Lambda}_\ell/n)^{-1}}^2$.

12:     Define $\rho^G(\mathcal{Y}(\underline{\mathcal{W}}_\ell)) := \min_{\mathbf{b}_{\underline{\mathbf{w}}}} \max_{\underline{\mathbf{w}},\underline{\mathbf{w}}' \in \underline{\mathcal{W}}_\ell} \|\underline{\mathbf{w}} - \underline{\mathbf{w}}'\|_{(\sum_{\underline{\mathbf{w}} \in \underline{\mathcal{W}}} \mathbf{b}_{\underline{\mathbf{w}}} \underline{\mathbf{w}}\, \underline{\mathbf{w}}^\top + \mathbf{\Lambda}_\ell/n)^{-1}}^2$.

13:     Set $\tau_\ell^G := \lceil \frac{64 B_*^\ell \rho^G(\mathcal{Y}(\mathcal{W}_\ell)) \log(4\ell^2|\mathcal{W}|/\delta_\ell)}{\epsilon_\ell^2} \rceil$. Then pull arm $\underline{\mathbf{w}} \in \underline{\mathcal{W}}$ exactly $\lceil \widehat{\mathbf{b}}_{\ell,\underline{\mathbf{w}}}^G \tau_\ell^G \rceil$ times and construct the least squares estimator $\widehat{\boldsymbol{\theta}}_\ell$ using only the observations of this phase where $\widehat{\boldsymbol{\theta}}_\ell$ is defined in (7). Note that $\widehat{\boldsymbol{\theta}}_\ell$ is also rotated following (4).

14:     Eliminate arms such that $\underline{\mathcal{W}}_{\ell+1} \leftarrow \underline{\mathcal{W}}_\ell \setminus \{\underline{\mathbf{w}} \in \underline{\mathcal{W}}_\ell : \max_{\underline{\mathbf{w}}' \in \underline{\mathcal{W}}_\ell} \langle \underline{\mathbf{w}}' - \underline{\mathbf{w}}, \widehat{\boldsymbol{\theta}}_\ell \rangle > 2\epsilon_\ell\}$

15:     $\ell \leftarrow \ell + 1$

16: Output the arm in $\underline{\mathcal{W}}_\ell$ and reshape to get the $\widehat{\mathbf{x}}_*$ and $\widehat{\mathbf{z}}_*$

---

### 2.3 Sample Complexity Analysis of Single-Task GOBLIN

We now analyze the sample complexity of GOBLIN in the single-task setting through the following theorem.

**Theorem 1.** *(informal)* *With probability at least* $1 - \delta$, *GOBLIN returns the best arms* $\mathbf{x}_*$, $\mathbf{z}_*$, *and the number of samples used is bounded by* $\widetilde{O}\left(\frac{(d_1+d_2)r}{\Delta^2} + \frac{\sqrt{d_1 d_2 r}}{S_r}\right)$.

**Discussion 1.** In Theorem 1 the first quantity is the number of samples needed to identify the best arms $\mathbf{x}_*$, $\mathbf{z}_*$ while the second quantity is the number of samples to learn $\boldsymbol{\Theta}_*$ (which is required to find the best arms). Note that the magnitude of $S_r$ would be free of $d_1, d_2$ since $\boldsymbol{\Theta}_*$ contains only $r$ nonzero singular values and $\|\boldsymbol{\Theta}_*\| \leq 1$, and hence we assume that $S_r = \Theta(1/\sqrt{r})$ (Kang et al., 2022). So the sample complexity of single-task GOBLIN scales as $\widetilde{O}(\frac{(d_1+d_2)r}{\Delta^2})$. However, if one runs RAGE (Fiez et al., 2019) on the arms in $\mathcal{X}, \mathcal{Z}$ then the sample complexity will scale as $\widetilde{O}(\frac{d_1 d_2}{\Delta^2})$.

**Proof (Overview) of Theorem 1: Step 1 (Subspace estimation in high dimension):** We denote the vectorized arms in high dimension as $\overline{\mathbf{w}} \in \overline{\mathcal{W}}$. We run the $E$-optimal design to sample the arms in $\overline{\mathcal{W}}$. Note that this $E$-optimal design satisfies the distribution assumption of Kang et al. (2022) which enables us to apply the Lemma 3 in Appendix A.1. This leads to $\|\widehat{\boldsymbol{\Theta}}_\ell - \boldsymbol{\Theta}_*\|_F^2 \leq \frac{C_1 d_1 d_2 r \log(2(d_1+d_2)/\delta)}{\tau_\ell^E}$ for some $C_1 > 0$. Also, note that in the first stage of the $\ell$-th phase by setting $\tau_\ell^E = \frac{\sqrt{8 d_1 d_2 r \log(4\ell^2 |\overline{\mathcal{W}}|/\delta_\ell)}}{S_r}$ and sampling each arm $\overline{\mathbf{w}} \in \overline{\mathcal{W}}$ exactly $\lceil \widehat{\mathbf{b}}_{\ell,\overline{\mathbf{w}}}^E \tau_\ell^E \rceil$ times we are guaranteed that $\|\boldsymbol{\theta}_{k+1:p}^*\|_2 = O(d_1 d_2 r/\tau_\ell^E)$. Summing up over $\ell = 1$ to $\lceil \log_2 \left(4\Delta^{-1}\right) \rceil$ we get that the total sample complexity of the first stage is bounded by $\widetilde{O}(\sqrt{d_1 d_2 r}/S_r)$.

**Step 2 (Effective dimension for rotated arms):** We rotate the arms $\overline{\mathbf{w}} \in \overline{\mathcal{W}}$ in high dimension to get the rotated arms $\underline{\mathbf{w}} \in \underline{\mathcal{W}}_\ell$ in step 10 of Algorithm 1. Then we show that the effective dimension of $\underline{\mathbf{w}}$ scales $8k \log\left(1 + \tau_{\ell-1}^G/\lambda\right)$ when $\lambda_\ell^\perp = \frac{\tau_{\ell-1}^G}{8k \log\left(1+\tau_{\ell-1}^G/\lambda\right)}$ in Lemma 7 of Appendix A.4. Note that this requires a different proof technique than Valko et al. (2014) where the budget $n$ is given apriori and effective dimension scales with $\log(n)$. This step also diverges from the pure exploration proof technique of Fiez et al. (2019); Katz-Samuels et al. (2020) as there is no parameter $\lambda_\ell^\perp$ to control during phase $\ell$, and the effective dimensions in those papers do not depend on phase length.

**Step 3 (Bounded Support):** For any phase $\ell$, we can show that $1 \leq \rho^G(\mathcal{Y}(\underline{\mathcal{W}}_\ell)) \leq p/\gamma_\mathcal{Y}^2$ where, $\gamma_\mathcal{Y} = \max\{c > 0 : c\mathcal{Y} \subset \text{conv}(\underline{\mathcal{W}} \cup -\underline{\mathcal{W}})\}$ is the gauge norm of $\mathcal{Y}$ (Rockafellar, 2015). Note that this is a worst-case dependence when $\rho^G(\mathcal{Y}(\underline{\mathcal{W}}_\ell))$ scales with $p$. Substituting this value of $\rho^G(\mathcal{Y}(\underline{\mathcal{W}}_\ell))$ in the definition of $\lambda_\ell^\perp$ we can show that $\boldsymbol{\Lambda}_\ell$ does not depend on $\underline{\mathbf{w}}$ or $\mathbf{y} = \underline{\mathbf{w}} - \underline{\mathbf{w}}'$. Then following Theorem 21.1 in Lattimore and Szepesvári (2020) we can show that the $G$-optimal design $\widehat{\mathbf{b}}_\ell^G$ is equivalent to $D$-optimal design $\widehat{\mathbf{b}}_\ell^D = \arg\max_{\mathbf{b}} \log \frac{\left|\sum_{\underline{\mathbf{w}} \in \underline{\mathcal{W}}_\ell} \mathbf{b}_{\underline{\mathbf{w}}} \underline{\mathbf{w}} \, \underline{\mathbf{w}}^\top + \boldsymbol{\Lambda}_\ell\right|}{|\boldsymbol{\Lambda}_\ell|}$. Then using Frank-Wolfe algorithm (Jamieson and Jain, 2022) we can show the support $\widehat{\mathbf{b}}_\ell^G$ or equivalently $\widehat{\mathbf{b}}_\ell^D$ is bounded by at most $\frac{8k \log(1+\tau_{\ell-1}^G/\lambda)(8k \log(1+\tau_{\ell-1}^G/\lambda)+1)}{2}$. This is shown in Lemma 9 (Appendix A.4).

**Step 4 (Phase length and Elimination):** Using the Lemma 9, concentration Lemma 5, and using the log determinant inequality in Lemma 7 and Proposition 1 (Appendix A.4) we show that the phase length in the second stage is given by $\tau_\ell^G = \lceil \frac{8B_*^\ell \rho(\mathcal{Y}(\underline{\mathcal{W}}_\ell)) \log(2|\mathcal{W}|/\delta)}{(\mathbf{x}^\top(\widehat{\boldsymbol{\theta}}_\ell - \boldsymbol{\theta}^*))^2} \rceil$. This is discussed in Discussion 3 (Appendix A.4). We show in Lemma 10 (Appendix A.4) that setting this phase length and sampling each active arm in $\underline{\mathcal{W}}_\ell$ exactly $\lceil \widehat{\mathbf{b}}_{\ell,\underline{\mathbf{w}}} \tau_\ell^G \rceil$ times results in the elimination of sub-optimal actions with high probability.

**Step 5 (Total Samples):** We first show that the total samples in the second phase are bounded by $O(\frac{k}{\gamma_\mathcal{Y}^2} \log(\frac{k \log_2(\Delta^{-1})|\mathcal{W}|}{\delta}) \lceil \log_2(\Delta^{-1}) \rceil)$ where the effective dimension $k = (d_1 + d_2)r$. Finally, we combine the total samples of phase $\ell$ as $(\tau_\ell^E + \tau_\ell^G)$. The final sample complexity is given by summing over all phases from $\ell = 1$ to $\lceil \log_2 \left(4\Delta^{-1}\right) \rceil$. The claim of the theorem follows by noting $\widetilde{O}(k/\gamma_\mathcal{Y}^2) \leq \widetilde{O}(k/\Delta^2)$.

## 3 Multi-task Representation Learning

In this section, we extend GOBLIN to multi-task representation learning for the bilinear bandit setting. In the multi-task setting, we now have $M$ tasks, where each task $m \in [M]$ has a reward model stated

in (1). The learning proceeds as follows: At each round $t = 1, 2, \cdots$, for each task $m \in [M]$, the learner selects a left and right action $\mathbf{x}_{m,t} \in \mathcal{X}$ and $\mathbf{z}_{m,t} \in \mathcal{Z}$. After the player commits the batch of actions for each task $\{\mathbf{x}_{m,t}, \mathbf{z}_{m,t} : m \in [M]\}$, it receives the batch of rewards $\{r_{m,t} : m \in [M]\}$. Finally recall that the goal is to identify the optimal left and right arms $\mathbf{x}_{m,*}, \mathbf{z}_{m,*}$ for each task $m$ with a minimum number of samples. We now state the following assumptions to enable representation learning across tasks.

**Assumption 1.** *(Low-rank Tasks)* *We assume that the hidden parameter $\mathbf{\Theta}_{m,*}$ for all the $m \in [M]$ have a decomposition $\mathbf{\Theta}_{m,*} = \mathbf{B}_1 \mathbf{S}_{m,*} \mathbf{B}_2^\top$ and each $\mathbf{S}_{m,*}$ has rank $r$.*

This is similar to the assumptions in Yang et al. (2020, 2022); Du et al. (2023) ensuring the feature extractors are shared across tasks in the bilinear bandit setting.

**Assumption 2.** *(Diverse Tasks)* *We assume that $\sigma_{\min}(\frac{1}{M} \sum_{m=1}^M \mathbf{\Theta}_{m,*}) \geq \frac{c_0}{S_r}$, for some $c_0 > 0$, $S_r$ is the $r$-th largest singular value of $\mathbf{\Theta}_{m,*}$ and $\sigma_{\min}(\mathbf{A})$ denotes the minimum eigenvalue of matrix $\mathbf{A}$.*

This assumption is similar to the diverse tasks assumption of Yang et al. (2020, 2022); Tripuraneni et al. (2021); Du et al. (2023) and ensures the possibility of recovering the feature extractors $\mathbf{B}_1$ and $\mathbf{B}_2$ shared across tasks.

Our extension of GOBLIN to the multi-task setting is now a phase-based, *three-stage* arm elimination algorithm. In GOBLIN each phase $\ell = 1, 2, \ldots$ consists of three stages; the stage for estimation of feature extractors $\mathbf{B}_1, \mathbf{B}_2$, which runs for $\tau_\ell^E$ rounds, the stage for estimation of $\mathbf{S}_{m,*}$ which runs for $\sum_m \widetilde{\tau}_{m,\ell}^E$ rounds, and a stage of pure exploration with rotated arms that runs for $\sum_m \tau_{m,\ell}^G$ rounds. We will define $\tau_{m,\ell}^E$ in Section 3.1, $\widetilde{\tau}_{m,\ell}^E$ in Section 3.2, while the rotated arms and $\tau_{m,\ell}^G$ are defined in Section 3.3. At the end of every phase, GOBLIN eliminates sub-optimal arms to build the active set for the next phase and stops when only the optimal left and right arms are remaining. Now we discuss the individual stages that occur at every phase $\ell = 1, 2, \ldots$ for multi-task GOBLIN.

## 3.1 Estimating Feature Extractors $\mathbf{B}_1$ and $\mathbf{B}_2$ (Stage 1 of Phase $\ell$)

In the first stage of phase $\ell$, GOBLIN leverages the batch of rewards $\{r_{m,t} : m \in [M]\}$ at every round $t$ from $M$ tasks to learn the feature extractors $\mathbf{B}_1$ and $\mathbf{B}_2$. To do this, GOBLIN first vectorizes the $\mathbf{x} \in \mathcal{X}, \mathbf{z} \in \mathcal{Z}$ into a new vector $\overline{\mathbf{w}} = [\mathbf{x}_{1:d_1}; \mathbf{z}_{1:d_2}] \in \overline{\mathcal{W}}_m$ and then solves the $E$-optimal design in step 3 of Algorithm 2. Similar to the single-task setting (Section 2) GOBLIN samples each $\overline{\mathbf{w}} \in \overline{\mathcal{W}}_m$ for $\lceil \tau_\ell^E \mathbf{b}_{\ell,\overline{\mathbf{w}}}^E \rceil$ times for each task $m$, where $\tau_\ell^E = \widetilde{O}(\sqrt{d_1 d_2 r}/S_r)$ and $\mathbf{b}_{\ell,\overline{\mathbf{w}}}^E$ is the solution to $E$-optimal design on $\overline{\mathbf{w}}$. Let the sampled arms for each task $m$ at round $s$ be denoted by $\mathbf{x}_{m,s}, \mathbf{z}_{m,s}$ which is obtained after reshaping $\overline{\mathbf{w}}_s$. Then it builds the estimator $\widehat{\mathbf{Z}}_\ell$ as follows:

$$\widehat{\mathbf{Z}}_\ell = \underset{\mathbf{\Theta} \in \mathbb{R}^{d_1 \times d_2}}{\arg\min} \ L_\ell(\mathbf{\Theta}) + \gamma_\ell \|\mathbf{\Theta}\|_{\mathrm{nuc}},$$

$$L_\ell(\mathbf{\Theta}) = \langle \mathbf{\Theta}, \mathbf{\Theta} \rangle - \frac{2}{M \tau_\ell^E} \sum_{m=1}^M \sum_{s=1}^{\tau_\ell^E} \langle \widetilde{\psi}_\nu(r_{m,s} \cdot Q(\mathbf{x}_{m,s} \mathbf{z}_{m,s}^\top)), \mathbf{\Theta} \rangle \tag{8}$$

Then it performs SVD decomposition on $\widehat{\mathbf{Z}}_\ell$, and let $\widehat{\mathbf{B}}_1, \widehat{\mathbf{B}}_2$ be the top-$k_1$ and top-$k_2$ left and right singular vectors of $\widehat{\mathbf{Z}}_\ell$ respectively. These are the estimation of the feature extractors $\mathbf{B}_1$ and $\mathbf{B}_2$.

## 3.2 Estimating Hidden Parameter $\mathbf{S}_{m,*}$ per Task (Stage 2 of phase $\ell$)

In the second stage of phase $\ell$, the goal is to recover the hidden parameter $\mathbf{S}_{m,*}$ for each task $m$. GOBLIN proceeds as follows: First, let $\widetilde{\mathbf{g}}_m = \mathbf{x}^\top \widehat{\mathbf{B}}_{1,\ell}$ and $\widetilde{\mathbf{v}}_m = \mathbf{z}^\top \widehat{\mathbf{B}}_{2,\ell}$ be the latent left and right arm respectively for each $m$. Then GOBLIN defines the vector $\widetilde{\mathbf{w}} = [\widetilde{\mathbf{g}}_m; \widetilde{\mathbf{v}}_m] \in \widetilde{\mathcal{W}}_m$ and then solves the $E$-optimal design in step 11 of Algorithm 2. It then samples for each task $m$, the latent arm $\widetilde{\mathbf{w}} \in \widetilde{\mathcal{W}}_m$ for $\lceil \widetilde{\tau}_{m,\ell}^E \widetilde{\mathbf{b}}_{m,\ell,\widetilde{\mathbf{w}}}^E \rceil$ times, where $\widetilde{\tau}_{m,\ell}^E := \widetilde{O}(\sqrt{k_1 k_2 r}/S_r)$ and $\widetilde{\mathbf{b}}_{m,\ell,\widetilde{\mathbf{w}}}^E$ is the solution to

$E$-optimal design on $\widetilde{\mathbf{w}}$. Then it builds estimator $\widehat{\mathbf{S}}_{m,\ell}$ for each task $m$ in step 12 as follows:

$$\widehat{\mathbf{S}}_{m,\ell} = \underset{\mathbf{\Theta} \in \mathbb{R}^{k_1 \times k_2}}{\arg\min}\ L'_\ell(\mathbf{\Theta}) + \gamma_\ell \|\mathbf{\Theta}\|_{\text{nuc}},$$

$$L'_\ell(\mathbf{\Theta}) = \langle \mathbf{\Theta}, \mathbf{\Theta} \rangle - \frac{2}{\widetilde{\tau}^E_{m,\ell}}\sum_{s=1}^{\widetilde{\tau}^E_{m,\ell}} \langle \widetilde{\psi}_\nu(r_{m,s} \cdot Q(\widetilde{\mathbf{g}}_{m,s}\widetilde{\mathbf{v}}^\top_{m,s})), \mathbf{\Theta} \rangle \tag{9}$$

Once GOBLIN recovers the $\widehat{\mathbf{S}}_{m,\ell}$ for each task $m$ it has reduced the $d_1 d_2$ bilinear bandit to a $k_1 k_2$ dimension bilinear bandit where the left and right arms are $\widetilde{\mathbf{g}}_m \in \mathcal{G}_m, \widetilde{\mathbf{v}}_m \in \mathcal{V}_m$ respectively.

### 3.3 Optimal Design for Rotated Arms per Task (Stage 3 of phase $\ell$)

In the third stage of phase $\ell$, similar to Algorithm 1, the multi-task GOBLIN defines the rotated arm set $\underline{\mathcal{G}}_m, \underline{\mathcal{V}}_m$ for each task $m$ for these $k_1 k_2$ dimensional bilinear bandits. Let the SVD of $\widehat{\mathbf{S}}_{m,\ell} = \widehat{\mathbf{U}}_{m,\ell}\widehat{\mathbf{D}}_{m,\ell}\widehat{\mathbf{V}}^\top_{m,\ell}$. Define $\widehat{\mathbf{H}}_{m,\ell} = [\widehat{\mathbf{U}}_{m,\ell}\widehat{\mathbf{U}}^\perp_{m,\ell}]^\top \widehat{\mathbf{S}}_{m,\ell}[\widehat{\mathbf{V}}_{m,\ell}\widehat{\mathbf{V}}^\perp_{m,\ell}]$. Then define the vectorized arm set so that the last $(k_1 - r) \cdot (k_2 - r)$ components are from the complementary subspaces as follows:

$$\underline{\mathcal{W}}_{m,\ell} = \big\{ \big[ \mathbf{vec}\,(\widetilde{\mathbf{g}}_{m,1:r}\widetilde{\mathbf{v}}^\top_{m,1:r})\,;\mathbf{vec}\,(\widetilde{\mathbf{g}}_{m,r+1:k_1}\widetilde{\mathbf{v}}^\top_{m,1:r})\,;\mathbf{vec}\,(\widetilde{\mathbf{g}}_{m,1:r}\widetilde{\mathbf{v}}^\top_{m,r+1:k_2})\,; $$
$$\mathbf{vec}\,(\widetilde{\mathbf{g}}_{m,r+1:k_1}\widetilde{\mathbf{v}}^\top_{m,r+1:k_2}) \big] \big\}$$

$$\widehat{\boldsymbol{\theta}}_{m,\ell,1:k} = [\mathbf{vec}(\widehat{\mathbf{H}}_{m,\ell,1:r,1:r}); \mathbf{vec}(\widehat{\mathbf{H}}_{m,\ell,r+1:k_1,1:r}); \mathbf{vec}(\widehat{\mathbf{H}}_{m,\ell,1:r,r+1:k_2})],$$
$$\boldsymbol{\theta}_{\ell,k+1:p} = \mathbf{vec}(\widehat{\mathbf{H}}_{m,\ell,r+1:k_1,r+1:k_2}). \tag{10}$$

This is shown in step 14 of Algorithm 2. Now we proceed similarly to Section 2.2. We construct a per-task optimal design for the rotated arm set $\underline{\mathcal{V}}_m, \underline{\mathcal{G}}_m$ and define the $\underline{\mathbf{w}} = [\underline{\widetilde{\mathbf{g}}}_{m,1:d_1}; \widetilde{\mathbf{v}}_{m,1:d_2}]$ and $\widetilde{\mathbf{w}} \in \widetilde{\mathcal{W}}_m$ where $\widetilde{\mathbf{g}}_m \in \underline{\mathcal{G}}_m$ and $\widetilde{\mathbf{v}}_m \in \underline{\mathcal{V}}_m$ respectively. Following (5) we know that to minimize the sample complexity for the $m$-th bilinear bandit we need to sample according to $G$-optimal design

$$\widehat{\mathbf{b}}^G_{m,\ell} = \underset{\mathbf{b}_{m,\underline{\mathbf{w}}}}{\arg\min}\ \max_{\underline{\mathbf{w}},\underline{\mathbf{w}}' \in \underline{\mathcal{W}}_{m,\ell}} \|\underline{\mathbf{w}} - \underline{\mathbf{w}}'\|^2_{(\sum_{\underline{\mathbf{w}} \in \underline{\mathcal{W}}_m} \mathbf{b}_{m,\underline{\mathbf{w}}}\underline{\mathbf{w}}\,\underline{\mathbf{w}}^\top + \mathbf{\Lambda}_{m,\ell}/n)^{-1}} \tag{11}$$

Then GOBLIN runs $G$-optimal design on the arm set $\underline{\mathcal{W}}_\ell$ following the (11) and then samples each $\underline{\mathbf{w}} \in \underline{\mathcal{W}}_{m,\ell}$ for $\lceil \tau^G_{m,\ell}\widehat{\mathbf{b}}^G_{m,\ell,\underline{\mathbf{w}}} \rceil$ times where $\widehat{\mathbf{b}}^G_{m,\ell,\underline{\mathbf{w}}}$ is the solution to the $G$-optimal design, and $\tau^G_\ell$ is defined in step 17 of Algorithm 2. So the total length of phase $\ell$, combining stages 1, 2 and 3 is $(\tau^E_\ell + \sum_m \widetilde{\tau}^E_{m,\ell} + \sum_m \tau^G_{m,\ell})$ rounds. Observe that the stage 1 and 2 design is on the whole arm set $\overline{\mathcal{W}}, \widetilde{\mathcal{W}}_m$ whereas the stage 3 design is on the refined active set $\underline{\mathcal{W}}_{m,\ell}$. Let at the stage 3 of $\ell$-th phase the actions sampled be denoted by the matrix $\underline{\mathbf{W}}_{m,\ell} \in \mathbb{R}^{\tau^G_{m,\ell} \times k_1 k_2}$ and observed rewards $\mathbf{r}_m \in \mathbb{R}^{\tau^G_{m,\ell} \times k_1 k_2}$. Define the positive diagonal matrix $\mathbf{\Lambda}_{m,\ell}$ according to (6) but set $p = k_1 k_2$ and $k = (k_1 + k_2)r$. Then similar to Section 2.2 we can build for each task $m$ only from the observations from this phase

$$\widehat{\boldsymbol{\theta}}_{m,\ell} = \arg\min_{\boldsymbol{\theta}} \tfrac{1}{2}\|\underline{\mathbf{W}}_{m,\ell}\boldsymbol{\theta} - \mathbf{r}_m\|^2_2 + \tfrac{1}{2}\|\boldsymbol{\theta}\|^2_{\mathbf{\Lambda}_{m,\ell}} \tag{12}$$

Finally GOBLIN eliminates the sub-optimal arms using the estimator $\widehat{\boldsymbol{\theta}}_{m,\ell}$ to build the next phase active set $\underline{\mathcal{W}}_{m,\ell}$ and stops when $|\underline{\mathcal{W}}_{m,\ell}| = 1$. The full pseudo-code is given in Algorithm 2.

### 3.4 Sample Complexity analysis of Multi-task GOBLIN

We now present the sample complexity of GOBLIN for the multi-task setting.

**Theorem 2.** *(informal)* With probability at least $1 - \delta$, GOBLIN returns the best arms $\mathbf{x}_{m,*}$, $\mathbf{z}_{m,*}$ for each task $m$, and the total number of samples is bounded by $\widetilde{O}\left( \frac{M(k_1+k_2)r}{\Delta^2} + \frac{M\sqrt{k_1 k_2 r}}{S_r} + \frac{\sqrt{d_1 d_2 r}}{S_r} \right)$.

**Discussion 2.** In Theorem 2 the first quantity is the sample complexity to identify the best arms $\mathbf{x}_{m,*}$, $\mathbf{z}_{m,*}$ and the second quantity is the number of samples to learn $\mathbf{S}_{m,*}$ for each task $m$. This is required to rotate the arms to reach the effective dimension of $(k_1 + k_2)r$. Finally, the third quantity is the number of samples needed to learn $\mathbf{\Theta}_{m,*}$ (which in turn is used to estimate the feature extractors $\mathbf{B}_1$ and $\mathbf{B}_2$ to learn the $\mathbf{S}_{m,*}$). Again we assume that $S_r = \Theta(1/\sqrt{r})$ (Kang et al., 2022). So the sample complexity of multi-task GOBLIN scales as $\widetilde{O}(M(k_1 + k_2)r/\Delta^2)$. However, if one runs DouExpDes (Du et al., 2023) then the sample complexity will scale as $\widetilde{O}(M(k_1 k_2)/\Delta^2)$ which is worse than GOBLINwhen $r \ll k_1$ or $k_2$.

---

**Algorithm 2** G-Optimal Design for Bilinear Bandits (GOBLIN) for multi-task setting

---

1: Input: arm set $\mathcal{X}, \mathcal{Z}$, confidence $\delta$, rank $r$ of $\boldsymbol{\Theta}_*$, spectral bound $S_r$ of $\boldsymbol{\Theta}_*$, $S, S_{m,\ell}^{\perp} = \frac{8k_1 k_2 r}{\widetilde{\tau}_{m,\ell}^E S_r^2} \log(\frac{k_1+k_2}{\delta_\ell})$, $\lambda, \lambda_{m,\ell}^{\perp} = \frac{\tau_{m,\ell-1}^G}{(8(k_1+k_2)r \log(1+\tau_{m,\ell-1}^G/\lambda))}$. Let $p = k_1 k_2$, $k = (k_1+k_2)r$.

2: Let $\underline{\mathcal{W}}_{m,1} \leftarrow \underline{\mathcal{W}}_m, \ell \leftarrow 1$, $\tau_0^G = \log(4\ell^2|\mathcal{X}|/\delta)$. Define $\boldsymbol{\Lambda}_{m,\ell}$ as in (6), $B_{m,*}^{\ell} := (8\sqrt{\lambda}S + \sqrt{\lambda_{m,\ell}^{\perp} S_{m,\ell}^{\perp}})$

3: Define arm $\overline{\mathbf{w}} = [\mathbf{x}_{1:d_1}; \mathbf{z}_{1:d_2}]$ and $\overline{\mathbf{w}} \in \overline{\mathcal{W}}_m$. Let $\tau_\ell^E = \frac{\sqrt{8d_1 d_2 r \log(4\ell^2|\mathcal{W}|/\delta_\ell)}}{S_r}$. Let $E$-optimal design be $\mathbf{b}_\ell^E = \arg\min_{\mathbf{b} \in \triangle_{\overline{\mathcal{W}}}} \|(\sum_{\overline{\mathbf{w}} \in \overline{\mathcal{W}}} b_{\overline{\mathbf{w}}} \overline{\mathbf{w}}\, \overline{\mathbf{w}}^\top)^{-1}\|$.

4: **while** $\exists m \in [M], |\underline{\mathcal{W}}_{m,\ell}| > 1$ **do**

5: $\quad \epsilon_\ell = 2^{-\ell}, \delta_\ell = \delta/\ell^2$.

6: $\quad$ **(Stage 1:) Explore the Low-Rank Subspace**

7: $\quad$ Pull arm $\overline{\mathbf{w}} \in \overline{\mathcal{W}}$ exactly $\lceil \widehat{\mathbf{b}}_{\ell,\overline{\mathbf{w}}}^E \tau_\ell^E \rceil$ times for each task $m$ and observe rewards $\{r_{m,t}\}_{t=1}^{\tau_\ell^E}$.

8: $\quad$ Compute $\widehat{\mathbf{Z}}_\ell$ using (8).

9: $\quad$ **(Stage 2:) Build $\widehat{\mathbf{S}}_{m,\ell}$ for each task $m$**

10: $\quad$ Let $\widehat{\mathbf{B}}_{1,\ell}, \widehat{\mathbf{B}}_{2,\ell}$ be the top-$k_1$ left and top-$k_2$ right singular vectors of $\widehat{\mathbf{Z}}_\ell$ respectively. Build $\widetilde{\mathbf{g}}_m = \mathbf{x}^\top \widehat{\mathbf{B}}_{1,\ell}$ and $\widetilde{\mathbf{v}}_m = \mathbf{z}^\top \widehat{\mathbf{B}}_{2,\ell}$ for all $\mathbf{x} \in \mathcal{X}$ and $\mathbf{z} \in \mathcal{Z}$ for each $m$.

11: $\quad$ Define a vectorized arm $\widetilde{\mathbf{w}} = [\widetilde{\mathbf{g}}_{m,1:k_1}; \widetilde{\mathbf{v}}_{m,1:k_2}]$ and $\widetilde{\mathbf{w}} \in \widetilde{\mathcal{W}}_m$ for each $m$. Let $\widetilde{\tau}_{m,\ell}^E = \frac{\sqrt{8k_1 k_2 r \log(4\ell^2|\mathcal{W}|/\delta_\ell)}}{S_r}$, and $\widetilde{\mathbf{b}}_{m,\ell}^E = \arg\min_{\mathbf{b}_m \in \triangle_{\widetilde{\mathcal{W}}_m}} \|(\sum_{\widetilde{\mathbf{w}} \in \widetilde{\mathcal{W}}_m} b_{m,\widetilde{\mathbf{w}}} \widetilde{\mathbf{w}}\, \widetilde{\mathbf{w}}^\top)^{-1}\|$.

12: $\quad$ Pull arm $\widetilde{\mathbf{w}} \in \widetilde{\mathcal{W}}_m$ exactly $\lceil \widetilde{\mathbf{b}}_{m,\ell,\widetilde{\mathbf{w}}}^E \widetilde{\tau}_{m,\ell}^E \rceil$ times and observe rewards $r_{m,t}$, for $t = 1, \ldots, \widetilde{\tau}_{m,\ell}^E$, for each task $m$. Then compute $\widehat{\mathbf{S}}_{m,\ell}$ using (9) for each $m$.

13: $\quad$ **(Stage 3:) Reduction to low dimensional linear bandits for each task $m$**

14: $\quad$ SVD of $\widehat{\mathbf{S}}_{m,\ell} = \widehat{\mathbf{U}}_{m,\ell} \widehat{\mathbf{D}}_{m,\ell} \widehat{\mathbf{V}}_{m,\ell}^\top$. Rotate arms in active set $\underline{\mathcal{W}}_{m,\ell-1}$ to build $\underline{\mathcal{W}}_{m,\ell}$ using (10).

15: $\quad$ Let $\widehat{\mathbf{b}}_{m,\ell}^G = \arg\min_{\mathbf{b}_{m,\underline{\mathbf{w}}}} \max_{\underline{\mathbf{w}},\underline{\mathbf{w}}' \in \underline{\mathcal{W}}_{m,\ell}} \|\underline{\mathbf{w}} - \underline{\mathbf{w}}'\|_{(\sum_{\underline{\mathbf{w}}_m \in \underline{\mathcal{W}}_m} b_{m,\underline{\mathbf{w}}} \underline{\mathbf{w}}_m \underline{\mathbf{w}}_m^\top + \boldsymbol{\Lambda}_{m,\ell}/n)^{-1}}^2$.

16: $\quad$ Define $\rho^G(\mathcal{Y}(\underline{\mathcal{W}}_{m,\ell})) = \min_{\mathbf{b}_{m,\underline{\mathbf{w}}}} \max_{\underline{\mathbf{w}},\underline{\mathbf{w}}' \in \underline{\mathcal{W}}_{m,\ell}} \|\underline{\mathbf{w}} - \underline{\mathbf{w}}'\|_{(\sum_{\underline{\mathbf{w}} \in \underline{\mathcal{W}}_m} b_{m,\underline{\mathbf{w}}} \underline{\mathbf{w}}\, \underline{\mathbf{w}}^\top + \frac{\boldsymbol{\Lambda}_{m,\ell}}{n})^{-1}}^2$.

17: $\quad$ Set $\tau_{m,\ell}^G = \frac{64 B_{m,*}^\ell \rho^G(\mathcal{Y}(\mathcal{W}_{m,\ell})) \log(4\ell^2|\mathcal{W}_m|/\delta_\ell)}{\epsilon_\ell^2}$. Then pull arm $\underline{\mathbf{w}} \in \underline{\mathcal{W}}_m$ for each task $m$ exactly $\lceil \widehat{\mathbf{b}}_{m,\ell,\underline{\mathbf{w}}} \tau_{m,\ell}^G \rceil$ times and construct the least squares estimator $\widehat{\boldsymbol{\theta}}_{m,\ell}$ using only the observations of this phase where $\widehat{\boldsymbol{\theta}}_{m,\ell}$ is defined in (12).

18: $\quad$ Eliminate arms such that $\underline{\mathcal{W}}_{m,\ell+1} \leftarrow \underline{\mathcal{W}}_{m,\ell} \setminus \left\{ \underline{\mathbf{w}}_m \in \underline{\mathcal{W}}_{m,\ell} : \max_{\underline{\mathbf{w}}_m' \in \underline{\mathcal{W}}_{m,\ell}} \left\langle \underline{\mathbf{w}}_m' - \underline{\mathbf{w}}_m, \widehat{\boldsymbol{\theta}}_{m,\ell} \right\rangle > 2\epsilon_{m,\ell} \right\}$

19: $\quad \ell \leftarrow \ell + 1$

20: Output the arm in $\underline{\mathcal{W}}_{m,\ell}$ and reshape to get the $\widehat{\mathbf{x}}_{m,*}$ and $\widehat{\mathbf{z}}_{m,*}$ for each task $m$.

---

**Proof (Overview) of Theorem 2: Step 1 (Subspace estimation in high dimension):** The first steps diverge from the proof technique of Theorem 1. We now build the average estimator $\widehat{\mathbf{Z}}_\ell$ to estimate the quantity $\mathbf{Z}_* = \frac{1}{M} \sum_{m=1}^M \boldsymbol{\Theta}_{*,m}$ using (8). This requires us to modify the Lemma 3 in Appendix A.1 and apply Stein's lemma (Lemma 1) to get a bound of $\|\widehat{\mathbf{Z}}_\ell - \mathbf{Z}_*\|_F^2 \leq \frac{C_1 d_1 d_2 r \log(2(d_1+d_2)/\delta)}{\tau_\ell^E}$ for some $C_1 > 0$. This is shown in Lemma 12 in Appendix A.6. Summing up over $\ell = 1$ to $\lceil \log_2(4\Delta^{-1}) \rceil$ we get that the total samples complexity of the first stage is bounded by $\widetilde{O}(\sqrt{d_1 d_2 r}/S_r)$.

**Step 2 (Estimation of left and right feature extractors):** Now using the estimator in (8) we get a good estimation of the feature extractors $\mathbf{B}_1$ and $\mathbf{B}_2$. Let $\widehat{\mathbf{B}}_{1,\ell}, \widehat{\mathbf{B}}_{2,\ell}$ be the top-$k_1$ left and top-$k_2$ right singular vectors of $\widehat{\mathbf{Z}}_\ell$ respectively. Then using the Davis-Kahan $\sin\theta$ Theorem (Bhatia, 2013) in Lemma 14, 15 (Appendix A.6) we have $\|(\widehat{\mathbf{B}}_{1,\ell}^{\perp})^\top \mathbf{B}_1\|, \|(\widehat{\mathbf{B}}_{2,\ell}^{\perp})^\top \mathbf{B}_2\| \leq \widetilde{O}(\sqrt{(d_1+d_2)r/M\tau_\ell^E})$.

**Step 3 (Estimation of $\widehat{\mathbf{S}}_{m,\ell}$ in low dimension):** Now we estimate the quantity $\widehat{\mathbf{S}}_{m,\ell} \in \mathbb{R}^{k_1 \times k_2}$ for each task $m$. To do this we first build the latent arms $\widetilde{\mathbf{g}}_m = \mathbf{x}^\top \widehat{\mathbf{U}}_\ell$ and $\widetilde{\mathbf{v}}_m = \mathbf{z}^\top \widehat{\mathbf{V}}_\ell$ for all $\mathbf{x} \in \mathcal{X}$

and $\mathbf{z} \in \mathcal{Z}$ for each $m$, and sample them following the $E$-optimal design in step 12 of Algorithm 2. We also show in Lemma 16 (Appendix A.6) that $\sigma_{\min}(\sum_{\widetilde{\mathbf{w}} \in \widetilde{\mathcal{W}}} \mathbf{b}_{\widetilde{\mathbf{w}}} \widetilde{\mathbf{w}} \widetilde{\mathbf{w}}^\top) > 0$ which enables us to sample following $E$-optimal design. Then use the estimator in (9). Then in Lemma 19 we show that $\|\widehat{\mathbf{S}}_{m,\ell} - \mu^* \mathbf{S}_{m,*}\|_F^2 \leq C_1 k_1 k_2 r \log\left(\frac{2(k_1+k_2)}{\delta_\ell}\right)/\tau_{m,\ell}^E$ holds with probability greater than $(1 - \delta)$. Also, note that in the second phase by setting $\widetilde{\tau}_{m,\ell}^E = \sqrt{8k_1 k_2 r \log(4\ell^2 |\mathcal{W}|/\delta_\ell)}/S_r$ and sampling each arm $\overline{\mathbf{w}} \in \overline{\mathcal{W}}$ exactly $\lceil \widehat{\mathbf{b}}_{\ell,\overline{\mathbf{w}}}^E \widetilde{\tau}_{m,\ell}^E \rceil$ times we are guaranteed that $\|\boldsymbol{\theta}_{k+1:p}^*\|_2 = O(k_1 k_2 r/\widetilde{\tau}_{m,\ell}^E)$ in the $\ell$-th phase. Summing up over $\ell = 1$ to $\lceil \log_2(4\Delta^{-1}) \rceil$ across each task $M$ we get that the total samples complexity of the second stage is bounded by $\widetilde{O}(M\sqrt{k_1 k_2 r}/S_r)$.

**Step 4 (Convert to $k_1 k_2$ bilinear bandits):** Once GOBLIN recovers $\widehat{\mathbf{S}}_{m,\tau_\ell^E}$ it rotates the arm set following (10) to build $\underline{\mathcal{W}}_m$ to get the $k_1 k_2$ bilinear bandits. The rest of the steps follow the same way as in steps 2, 3 and 4 of proof of Theorem 1.

**Step 5 (Total Samples):** We show the total samples in the third phase are bounded by $O(\frac{k}{\gamma_{\mathcal{Y}}^2} \log(\frac{k \log_2(\Delta^{-1})|\mathcal{W}|}{\delta}) \lceil \log_2(\Delta^{-1}) \rceil)$ where the effective dimension $k = (k_1 + k_2)r$. The total samples of phase $\ell$ is given by $\tau_\ell^E + \sum_m (\widetilde{\tau}_{m,\ell}^E + \tau_{m,\ell}^G)$. Finally, we get the total sample complexity by summing over all phases from $\ell = 1$ to $\lceil \log_2(4\Delta^{-1}) \rceil$. The claim of the theorem follows by noting $\widetilde{O}(k/\gamma_{\mathcal{Y}}^2) \leq \widetilde{O}(k/\Delta^2)$.

## 4 Experiments

In this section, we conduct proof-of-concept experiments on both single and multi-task bilinear bandits. In the single-task experiment, we compare against the state-of-the-art RAGE algorithm (Fiez et al., 2019). We show in Figure 1 (left) that GOBLIN requires fewer samples than the RAGE with an increasing number of arms. In the multi-task experiment, we compare against the state-of-the-art DouExpDes algorithm (Du et al., 2023). We show in Figure 1 (right) that GOBLIN requires fewer samples than DouExpDes with an increasing number of tasks. As experiments are not a central contribution, we defer a fuller description of the experimental set-up to Appendix A.8.

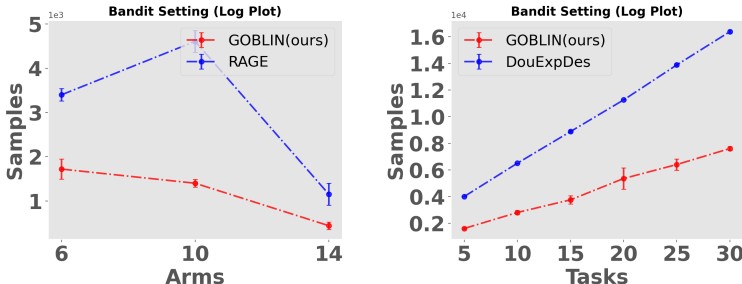

Figure 1: (Left) Single-task experiment: results show the number of samples required to identify the optimal action pair for differing numbers of actions. (Right) Multi-task experiment: results show the number of samples required to identify the optimal action pair for varying numbers of tasks.

## 5 Conclusions and Future Directions

In this paper, we formulated the first pure exploration multi-task representation learning problem. We introduce an algorithm, GOBLIN that achieves a sample complexity bound of $\widetilde{O}((d_1 + d_2)r/\Delta^2)$ which improves upon the $\widetilde{O}((d_1 d_2)/\Delta^2)$ sample complexity of RAGE (Fiez et al., 2019) in a single-task setting. We then extend GOBLIN for multi-task pure exploration bilinear bandit problems by learning latent features which enables sample complexity that scales as $\widetilde{O}(M(k_1 + k_2)r/\Delta^2)$ which improves over the $\widetilde{O}(M(k_1 k_2)/\Delta^2)$ sample complexity of DouExpDes (Du et al., 2023). Our analysis opens an exciting opportunity to analyze representation learning in the kernel and neural bandits (Zhu et al., 2021; Mason et al., 2021). We can leverage the fact that this type of optimal design does not require the arm set to be an ellipsoid (Du et al., 2023) which enables us to extend our analysis to non-linear representations.

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
