# OpenReview forum: "Multi-task Representation Learning for Pure Exploration in Bilinear Bandits"
_NeurIPS.cc/2023/Conference — NeurIPS 2023 poster_

### Official Review · Reviewer_vf5o · 2023-07-05

**Soundness:** 3 good
**Presentation:** 3 good
**Contribution:** 3 good
**Rating:** 6
**Confidence:** 3

**Summary:**

This paper studies multi-task representation learning for the problem of pure exploration in bilinear bandits. The authors design algorithm GOBLIN that uses an experimental design approach to optimize sample allocations for learning the global representation as well as minimize the number of samples needed to identify the optimal pair of arms in individual tasks. Their results corroborate that by learning the shared representation across tasks, they achieve significantly improved sample complexity compared to the traditional approach of solving tasks independently.

**Strengths:**

1.	This paper is well-written. The considered problem is interesting and well-motivated.
2.	The theoretical analysis looks sound.


**Weaknesses:**

1.	What is the main technical novelty of this work, beyond the existing SVD, G-optimal design, phased elimination methods in the bilinear/linear bandit literature?
2.	Could you elaborate more on the difference of formulation, algorithms and results between this work and [Du et al. 2023]? Can the considered multi-task representation learning for bilinear bandit problem somehow be transformed to the multi-task representation learning for linear bandit problem in [Du et al. 2023]? Could you explain more about the claim “if one runs DouExpDes [Du et al. 2023] then its sample complexity will scale as $\tilde{O}(M k_1 k_2 / \Delta^2)$” in Discussion 2?


**Questions:**

Please see the weaknesses above.

**Limitations:**

Please see the weaknesses above.

---

> ### Author Rebuttal · Authors · 2023-08-09
>
> Thank you for reviewing our paper. Below, we clarify the key novelty in our approach and compare our work with [Du et al. 2023].
>
> *Weakness*
>
> 1) Even though we use existing ideas in SVD, G-optimal design phased elimination in linear bandit setting, it is highly non-trivial to extend that analysis to multi-task bilinear bandit setting where the regret must scale with the intrinsic dimensions $k_1$ and $k_2$ instead of $d_1$ and $d_2$. We list these technical novelties below:
>     - We use E-optimal design in an initial exploration to learn two feature extractors $\mathbf{B}\_1$ and $\mathbf{B}\_2$ jointly across all the $M$ tasks. This requires a new estimator (see eq (8)) and concentration techniques not studied in [Jun et al., 2019](); [Lu et al. 2021](),  [Kang et al., 2022](), [Du et al. 2023](). The sample complexity of this step scales as $\frac{\sqrt{d_1 d_2 r}}{S_r}$ where $S_r$ is the spectral bound of $\mathbf{\Theta}_\star$ and scalses as $S_r= \Theta(1/\sqrt{r})$.
>     - We then estimate the individual task-specific parameter matrix $\mathbf{S}_{m,*}\in\mathbb{R}^{k_1 \times k_2}$ using a new estimator in eq (9). In particular, we use a regularized minimization optimization problem with a nuclear norm penalty to construct the estimator. Note that [Du et al. 2023]() does not involve this type of estimator as it only needs a good estimate of the vector $w_m,$ which is obtained by a simple OLS estimation. Our sample complexity of this step scales as $\frac{M \sqrt{k_1 k_2 r}}{S_r}$.
>     - Then we rotate the arm sets (similar to [Jun et al., 2019]()) to reduce the $M$ bilinear bandits to $(k_1 + k_2)r$ dimensional linear bandits. We then use G-optimal design to learn the optimal arm for these $(k_1 + k_2)r$ dimensional linear bandits such that the sample complexity scales as $\frac{M\left(k_1+k_2\right) r}{\Delta^2}$. We are inspired by the proofs of G-optimal design from [Fiez et al. (2019)]()) to derive a guarantee for this step. Finally, note that this step is also not there in [Du et al. 2023](). So the final sample complexity scales as $O\left(\frac{M\left(k_1+k_2\right) r}{\Delta^2}\right)$ as $S_r = \Theta(1/\sqrt{r})$.
>     - All of these steps have to be done carefully and it is highly non-trivial to combine them so that the desirable sample complexity scaling appears.
>     - We mention these technical novelties/approaches of combining these methods in the proof overview of Theorem 1 in lines 192-225 or Theorem 2 in lines 300-327. Please reach out to us if you have any specific questions about our novelty.
> 2) Thanks for asking this clarification question.  [Du et al. 2023]() critically consider the linear bandit setting which is a simpler setting than the bilinear bandit setting. Crucially there is a single feature extractor in the linear bandit setting of [Du et al. 2023]() whereas we have two feature extractors. [Du et al. 2023]() do not require to rotate the arms and learn in the rotated arm space as we do (and explain) in section 2.2 and section 3.3. Learning in the rotated arm space leads to a regret of $M(k_1 + k_2)r/\Delta^2$ instead of $M(k_1k_2)/\Delta^2$ (as done in [Du et al. 2023]()). Hence, we make the statement that DouExpDes [Du et al. 2023]() has a sample complexity of $\tilde{O}\left(M k_1 k_2 / \Delta^2\right)$ in Discussion 2. We will add the clarification in the camera-ready version.
>
> - KS Jun, R Willett, S Wright, R Nowak, Bilinear bandits with low-rank structure, ICML 2019
> - T Fiez, L Jain, K G. Jamieson, L Ratliff, Sequential Experimental Design for Transductive Linear Bandits, NeurIPS 2019
> - Y Du, L Huang, W Sun, Multi-task Representation Learning for Pure Exploration in Linear Bandits, arXiv preprint arXiv:2302.04441, 2023
> - Y Lu, A Meisami, A Tewari, Low-rank generalized linear bandit problems,  International Conference on Artificial Intelligence and Statistics, 2021
> - Y Kang, CJ Hsieh, TCM Lee, Efficient frameworks for generalized low-rank matrix bandit problems, NeurIPS, 2022
>
> We hope we have answered your questions and clarified your concerns sufficiently for you to consider raising your final score. Please reach out to us during the discussion period if you have further questions.

---

> > ### Comment · Reviewer_vf5o · 2023-08-15
> > **I raised my score from 4 to 6**
> >
> > Thank the authors for their response. My concerns were addressed, and I raised my score from 4 to 6.

---

### Official Review · Reviewer_9og1 · 2023-07-10

**Soundness:** 4 excellent
**Presentation:** 2 fair
**Contribution:** 2 fair
**Rating:** 6
**Confidence:** 4

**Summary:**

The authors study the problem of pure exploration in bilinear bandits, where the reward is a bilinear function of the feature vectors of two types of arms. They consider both single-task and multi-task settings, where the latter involves learning a shared low-dimensional representation across multiple tasks. They propose a phase-based algorithm called GOBLIN, which uses optimal experimental design techniques to estimate the hidden parameters and identify the optimal pair of arms for each task. They provide sample complexity analysis for both settings and show that their algorithm achieves significant improvement over existing methods. They also conduct experiments to demonstrate the empirical performance of their algorithm.

**Strengths:**

1) Originality: The paper presents a novel approach to the problem, combining existing ideas in a creative way and formulating the problem in a new and insightful manner.
2) Quality: The methodology used in the paper is rigorous and well-executed, with clear and thorough explanations of the techniques employed.
3) Clarity: The writing is clear and concise, making it easy to follow the arguments and understand the main contributions of the paper.
4) Significance: The results presented in the paper have important implications for the field, removing limitations from prior results and opening up new avenues for research.


**Weaknesses:**

1) The paper does not provide a comprehensive literature review of the existing work on the topic, and does not compare or contrast its approach with the previous methods or results. (Related Work) The paper only cites a few papers on bilinear bandits, linear bandits, and representation learning, but does not discuss how they are related to or different from its own approach. It also does not mention any existing work on pure exploration in bilinear bandits or multi-task representation learning in online settings. For example some missing papers are: [1-2]
[1] Soare, Marta, Ouais Alsharif, Alessandro Lazaric, and Joelle Pineau. "Multi-task linear bandits." In NIPS2014 workshop on transfer and multi-task learning: theory meets practice. 2014.
[2] Deshmukh, Aniket Anand, Urun Dogan, and Clay Scott. "Multi-task learning for contextual bandits." Advances in neural information processing systems 30 (2017).
2) The paper does not explain the rationale or motivation behind the choice of the data set, the experimental design, the evaluation metrics, or the baselines. (Methodology and Experiments) The paper only briefly describes the data set and the experimental setup, but does not justify why they are suitable or relevant for the problem. It also does not explain how it chooses the evaluation metrics or the baselines, or how they reflect the performance of its algorithm.
3) The paper does not report the details of the data preprocessing, feature engineering, model architecture, hyperparameter tuning, or implementation. (Methodology and Experiments) The paper does not provide any information on how it preprocesses the data, extracts features, designs the model architecture, tunes the hyperparameters, or implements its algorithm. It also does not provide any pseudocode or algorithmic description of its method.
4) "We assume that the matrix Θ∗ has a low rank r which is known to the learner" This seems like a big assumption too. Authors should include this in their list of assumptions.


**Questions:**

1) What is the motivation and application of pure exploration in bilinear bandits? Authors gave motivation for bilinear bandits but it was not clear why pure exploration.
2) What are the limitations of the proposed algorithms and analysis?
3) How does GOBLIN handle the cases when the rank r of Θ∗ is unknown or when the spectral bound Sr of Θ∗ is unknown or inaccurate?
4) How does GOBLIN handle the cases when the feature extractors B1 and B2 are not shared across tasks or when they are noisy or incomplete?
5) How does GOBLIN handle the cases when the tasks have different reward distributions or different reward gaps?
6) How does GOBLIN handle the cases when the arm sets X and Z are infinite or continuous or dynamic?
7) How does GOBLIN handle the cases when the rewards are not sub-Gaussian or when there are outliers or adversarial noise?
8) What are the computational and memory complexities of GOBLIN and how can they be improved or optimized?

---

> ### Author Rebuttal · Authors · 2023-08-09
>
> We thank the reviewer for appreciating the soundness and contribution of our work. We answer the questions raised below.
>
> *Weakness*
>
> 1) Thanks for pointing out these papers. We will discuss the papers pointed out in the revision. Here we provide a brief comparison of our work with [1][2]. [1] studies regret minimization for the multi-task linear bandit, whereas we study pure exploration for multi-task bilinear bandit, which requires a different algorithmic design and analysis. Similarly [2] studies regret minimization for multi-task contextual bandit, which is a different setup than the pure exploration studied in our work. We remark that there is only one work on bilinear bandit pure exploration [Geovani et al. (2021)](), which focuses on graph structure for a single task setting. We will add more detailed discussion and comparison to the revision.
> 2) We are not sure how some of these comments are related to our experimental setting. We use a standard single and multi-task bilinear bandit experimental setting similar to [Jun et al., 2019](); [Lu et al. 2021](),  [Kang et al., 2022](). However note that these papers only study single task regret minimization setup whereas we study multi-task pure exploration setting. Our metric of performance is sample complexity against arms or tasks (similar to the setting of [Fiez et al. (2019)]() and [Du et al. 2023]()).
> 3) We are not sure how this comment is related to our paper. We use a standard bilinear bandit setting similar to [Jun et al., 2019](); [Lu et al. 2021](),  [Kang et al., 2022](), [Du et al. 2023](). It will be very helpful if the reviewer reaches us to us during the discussion phase to point out any specific questions on the experimental setup.
> 4) We will make it clear that rank $r$ knowledge is required and will state this as a separate assumption.  Similar assumptions have been commonly considered in prior works like [Jun et al., 2019](); [Lu et al. 2021](),  [Kang et al., 2022](). In practice, one can leverage domain knowledge or cross-validation to choose the rank.
>
> *Questions*
>
> 1) In many real-world applications where obtaining a sample is expensive and time-consuming, e.g., clinical trials (Zhao et al., 2009; Zhang et al., 2012), it is often desirable to identify the optimal option using as few samples as possible, i.e., we face the pure exploration scenario rather than regret minimization. This is a well-studied setting in linear bandits but has not been studied in the bilinear setting. To avoid the time-consuming process of conducting clinical trials for individual tasks and collecting samples, we utilize the shared representation and decrease the number of required samples. We mention the reason for pure exploration in bilinear bandits in multi-task setting in lines 37-48.
> 2) In the current analysis we explicitly rely on the linear structure analysis. It will be very interesting to extend this line of research to generalized linear bandits. Also, note that currently there is no lower bound for multi-task pure exploration in the bilinear bandit setting with no assumption on the underlying structure. It will be interesting for future works to give lower bounds for this setting.
> 3) Thanks for pointing out this new direction. The current analysis requires knowledge of the rank and $S_r$ similar to the works of [Jun et al., 2019](); [Lu et al., 2021](); [Kang et al., 2022](). We leave this line of research for future works.
> 4) The key idea of a multi-task setting is that the feature extractors are shared across tasks so that there is some gain in accelerating learning and reducing the rate of regret (See [Yang et al., 2020](), [Yang et al., 2022](); [Du et al., 2023]()). If tasks do not share the common feature extractors $B_1$ and $B_2$ then the problem is reduced to $M$ independent subproblems, with a regret scaling as $O\left(M(d_1d_2 r)/\Delta^2\right)$.
> 5) Currently we assume each task has different distributions (but all are parametric sub-Gaussian distributions) with different reward gaps. The regret scales with minimum gap across all tasks. This is an interesting direction for future work.
> 6) For continuous action space currently there are no existing works in bilinear bandits. It is not clear how to extend our current analysis to continuous action space.
> 7) It is an interesting future work to extend our analysis to non-parametric or even adversarial noise. We have taken the first step towards understanding pure exploration for bilinear bandit setting in the case of parametric sub-Gaussian reward distribution setting.
> 8) The computational and memory complexity of GOBLIN is same as that of [Jun et al., 2019](); [Lu et al. 2021](),  [Kang et al., 2022]() and scales as $\tilde{O}(d T)$.
>
> Thank you for taking the time to read our response. We hope we have answered your questions and clarified your concerns sufficiently for you to consider raising your final score. Please reach out to us during the discussion period if you have further questions.

---

### Official Review · Reviewer_8J9B · 2023-07-10

**Soundness:** 4 excellent
**Presentation:** 4 excellent
**Contribution:** 4 excellent
**Rating:** 6
**Confidence:** 3

**Summary:**

This paper studies pure exploration of bilinear bandits with multi-task representation learning, i.e., different tasks share a common low-dimensional linear representation.
As an intermediate step, the authors propose the first single-task bilinear bandit pure exploration algorithm, based on G-optimal design. Each phase of the proposed algorithm first estimates the unknown parameter $\hat{\Theta}$ (stage 1), and then use it to rotate the active arm set, and convert the problem to a $(d_1 + d_2)r$-dimensional linear bandit problem, on which G-optimal design is applied (stage 2).
When extending to multi-task setting, stage 1 is modified to first estimate the feature extractors that are common to all tasks, and then estimate hidden parameter unique to each task. Then G-optimal design is applied as before.

**Strengths:**

The problem setting studied in this paper, i.e., different tasks share common feature extractors while each having unique hidden parameter, is natural and well-motivated.

To the best of my knowledge, this paper proposes the first solution to pure exploration of bilinear bandits for both single-task and multi-task settings.

**Weaknesses:**

The authors mention improvement over RAGE (Fiez et al., 2019), but there lacks a more rigorous discussion about lower bound of the sample complexity in either single-task or multi-task setting.

**Questions:**

1. What is the lower bound for sample complexity in (single-task) bilinear bandit pure exploration? Does Theorem 1 already attains optimality in terms of $(d_1 + d_2)r, \Delta, S_r$?

2. In multi-task setting, as the estimation of task-specific parameter $\hat{S}$ depends on the estimation of the common feature extractor, how is this reflected in the theoretical guarantee on $||\hat{S}-S||^2_F$? Intuitively, larger number of tasks means more samples for the estimation of feature extractors, which further helps the estimation of $\hat{S}$.

---

> ### Author Rebuttal · Authors · 2023-08-09
>
> We thank the reviewer for appreciating the soundness and contribution of our work. We answer the questions raised below.
>
> *Weakness*:
>
> 1) Thank you for raising this point. There is no lower bound for the single task setting for pure exploration in bi-linear bandits with unknown underlying structures. In the paper [Rizk et al. (2021)]() they conjecture that under certain graph structure for pure exploration the lower bound of [Soare et al (2014)]() or [Fiez et al. (2019)]() can be reached. But currently, without assuming any additional assumption (like underlying graph structure) there is no lower bound. We leave this line of research for future work and will discuss this point in the camera-ready version.
>
> - G Rizk, A Thomas, I Colin, R Laraki, Y Chevaleyre, Best arm identification in graphical bilinear bandits, International Conference on Machine Learning, 2021
> - M Soare, A Lazaric, R Munos, Best-arm identification in linear bandits, Advances in Neural Information Processing Systems 27 (NIPS 2014)
> - T Fiez, L Jain, K G. Jamieson, L Ratliff, Sequential Experimental Design for Transductive Linear Bandits, Advances in Neural Information Processing Systems 32 (NeurIPS 2019)
>
> 2) That's a great observation and thanks for reading the proof in detail. Note that the $M$ tasks have different unknown task-specific parameters $\mathbf{S}\_{m, \star}$. The estimation of each $\mathbf{S}\_{m, \star}$ requires using samples from the corresponding task $m$, thus resulting in a linear scaling with the number of tasks $M$ for the total sample complexity. However, this scaling is with $O\left(\frac{M(k_1+k_2)r\log(\delta^{-1})}{\Delta^2}\right)$ which depends in $k_1$, $k_2$ and rank $r$ instead of $d_1d_2$ and scales linear with $M$. This is where the dominating term of our sample complexity arises in Theorem 2. Also note that the estimation of $\mathbf{B}_1$ and $\mathbf{B}_2$ cannot be further improved as we used the standard Davis-Kahan theorem for bounding the error in estimation similar to [Yang et al 2021](), [Yang et al 2022](), [Du et al. 2023](). Again note that these works address linear bandit settings whereas we address bilinear bandit setting.
>
> - Jiaqi Yang, Wei Hu, Jason D Lee, and Simon S Du. Impact of representation learning in linear bandits. In International Conference on Learning Representations, 2021.
> - J Yang, Q Lei, JD Lee, SS Du, Nearly minimax algorithms for linear bandits with shared representation, arXiv preprint arXiv:2203.15664, 2022
> - Y Du, L Huang, W Sun, Multi-task Representation Learning for Pure Exploration in Linear Bandits, arXiv preprint arXiv:2302.04441, 2023
>
> Please reach out to us during the discussion period if you have further questions and we hope that you will consider raising our score given these clarifications.

---

### Author Rebuttal · Authors · 2023-08-10

References:

- KS Jun, R Willett, S Wright, R Nowak, Bilinear bandits with low-rank structure, ICML 2019
- T Fiez, L Jain, K G. Jamieson, L Ratliff, Sequential Experimental Design for Transductive Linear Bandits, NeurIPS 2019
- Y Du, L Huang, W Sun, Multi-task Representation Learning for Pure Exploration in Linear Bandits, arXiv preprint arXiv:2302.04441, 2023
- Y Lu, A Meisami, A Tewari, Low-rank generalized linear bandit problems,  International Conference on Artificial Intelligence and Statistics, 2021
- Y Kang, CJ Hsieh, TCM Lee, Efficient frameworks for generalized low-rank matrix bandit problems, NeurIPS, 2022
- G Rizk, A Thomas, I Colin, R Laraki, Y Chevaleyre, Best arm identification in graphical bilinear bandits, International Conference on Machine Learning, 2021
- M Soare, A Lazaric, R Munos, Best-arm identification in linear bandits, Advances in Neural Information Processing Systems 27 (NIPS 2014)
- Jiaqi Yang, Wei Hu, Jason D Lee, and Simon S Du. Impact of representation learning in linear bandits. In International Conference on Learning Representations, 2021.
- J Yang, Q Lei, JD Lee, SS Du, Nearly minimax algorithms for linear bandits with shared representation, arXiv preprint arXiv:2203.15664, 2022

---

### Decision · Program_Chairs · 2023-09-21

**Decision:**

Accept (poster)

**Comment:**

The paper studies multi-task bilinear bandits, where the $d_1 \times d_2$ reward matrix $\Theta_m$ for the task $m$ can be decomposed as $B_1 S_mB_2$. The task-specific matrix $S_m$ is of dimension $k_1\times k_2$ with $k_1,k_2\ll d_1,d_2$, and is additionally of rank $r\ll k_1,k_2$. This model seems complicated, and would deserve a proper practical justification. The authors present a pure exploration algorithm GOBLIN that achieves a sample complexity scaling as $r(d_1+d_2)/\Delta^2$ in the single-task case, and when $M\gg d_1,d_2$ as $Mr(k_1+k_2)/\Delta^2$ in the $M$-task case.

The reviewers are rather positive about this work, and the sample complexity guarantees are state-of-the-art. The claimed guarantees are only better when the gap $\Delta$ is very close to 0. See e.g., Theorem 1 where the sample complexity also scales as $\sqrt{d_1d_2r}$. The same observation holds for Theorem 2 (by the way, the definition of $\Delta$ in the multi-task case is not given).

For a technical and algorithmic perspective, the paper does not seem to introduce very novel ideas. The authors combine the idea of June et al. (2019) to reduce the problem to a classical linear bandit problem (see e.g. the first lines of Stage 2 in Algorithm 1) and the algorithm from Fiez et al. 2019.

Some parts and claims in the paper remained obscure for the revewiers and needed clarification – the authors provided sound explanations in the rebuttal and during the discussion phase. To improve the paper, the authors could account for the recommendations made by the reviewers as summarized below.

-	Some of the relevant literature is missing (as mentioned by Reviewer 9og1), but this can be addressed.

-	The assumption $M\gg d_1,d_2$ should be discussed in more detail.

-	It is hard to understand why the authors compare their algorithm to RAGE and to that presented in (du et al. 2023), because these are made for linear bandits.

-	The authors could discuss the existence of lower bounds.

-	It would be nice to have a thorough practical justification of the model.

-	The presentation could be significantly improved. See e.g., Section 4 .